# Single Ultra-High Dose Rate Proton Transmission Beam for Whole Breast FLASH-Irradiation: Quantification of FLASH-Dose and Relation with Beam Parameters

**DOI:** 10.3390/cancers15092579

**Published:** 2023-04-30

**Authors:** Patricia van Marlen, Steven van de Water, Max Dahele, Berend J. Slotman, Wilko F. A. R. Verbakel

**Affiliations:** Department of Radiation Oncology, Amsterdam UMC, Vrije Universiteit Amsterdam, Cancer Center Amsterdam, De Boelelaan 1117, 1118, 1081 HV Amsterdam, The Netherlands

**Keywords:** whole breast irradiation, proton transmission beams, FLASH, ultra-high dose-rate

## Abstract

**Simple Summary:**

Ultra-high dose-rate (UHDR) irradiation can lead to a FLASH-effect which reduces the biological effect on healthy tissue without affecting the dose to the tumor. UHDRs can currently be achieved using 250 MeV (transmission) proton beams. Up until now, suggested treatment sites potentially benefiting from a FLASH effect include, e.g., small lung, skin, and brain tumors. We suggest the use of FLASH for breast tumors because almost the entire target consists of healthy tissue and hypofractionated dose schedules are common. In this study, we demonstrate that whole breast irradiation (WBI) can be performed using a single transmission proton beam. We evaluated the necessary proton beam parameters in order to achieve the maximum FLASH-effect according to our current knowledge. Although currently not clinically applicable, we hypothesize that a potential FLASH-effect can even allow for further hypofractionation of WBI.

**Abstract:**

Healthy tissue-sparing effects of FLASH (≥40 Gy/s, ≥4–8 Gy/fraction) radiotherapy (RT) make it potentially useful for whole breast irradiation (WBI), since there is often a lot of normal tissue within the planning target volume (PTV). We investigated WBI plan quality and determined FLASH-dose for various machine settings using ultra-high dose rate (UHDR) proton transmission beams (TBs). While five-fraction WBI is commonplace, a potential FLASH-effect might facilitate shorter treatments, so hypothetical 2- and 1-fraction schedules were also analyzed. Using one tangential 250 MeV TB delivering 5 × 5.7 Gy, 2 × 9.74 Gy or 1 × 14.32 Gy, we evaluated: (1) spots with equal monitor units (MUs) in a uniform square grid with variable spacing; (2) spot MUs optimized with a minimum MU-threshold; and (3) splitting the optimized TB into two sub-beams: one delivering spots above an MU-threshold, i.e., at UHDRs; the other delivering the remaining spots necessary to improve plan quality. Scenarios 1–3 were planned for a test case, and scenario 3 was also planned for three other patients. Dose rates were calculated using the pencil beam scanning dose rate and the sliding-window dose rate. Various machine parameters were considered: minimum spot irradiation time (minST): 2 ms/1 ms/0.5 ms; maximum nozzle current (maxN): 200 nA/400 nA/800 nA; two gantry-current (GC) techniques: energy-layer and spot-based. For the test case (PTV = 819 cc) we found: (1) a 7 mm grid achieved the best balance between plan quality and FLASH-dose for equal-MU spots; (2) near the target boundary, lower-MU spots are necessary for homogeneity but decrease FLASH-dose; (3) the non-split beam achieved >95% FLASH for favorable (not clinically available) machine parameters (SB GC, low minST, high maxN), but <5% for clinically available settings (EB GC, minST = 2 ms, maxN = 200 nA); and (4) splitting gave better plan quality and higher FLASH-dose (~50%) for available settings. The clinical cases achieved ~50% (PTV = 1047 cc) or >95% (PTV = 477/677 cc) FLASH after splitting. A single UHDR-TB for WBI can achieve acceptable plan quality. Current machine parameters limit FLASH-dose, which can be partially overcome using beam-splitting. WBI FLASH-RT is technically feasible.

## 1. Introduction

There is great interest in FLASH-radiotherapy (RT), which involves irradiation at ultra-high dose rates (UHDRs, e.g., ≥40 Gy/s). UHDR irradiation seems to cause less normal tissue damage while maintaining a similar anti-tumor effect to conventional dose-rate radiotherapy. This so-called FLASH-effect has been observed in various in vitro [1] and in vivo models, including mouse lung [2,3,4], brain [5,6,7,8], abdominal tissues [6,9,10], and mouse, pig, cat, and canine skin [11,12,13,14]. Observations have been published in a human patient with a cutaneous lymphoma [15], and two clinical trials treating bone or melanoma metastases are currently in progress [16,17].

FLASH-RT is therefore of interest for potential sparing of organs-at-risk (OARs) inside and outside the planning target volume (PTV). The PTV is a concept defined for treatment planning and includes the target region (e.g., a macroscopic tumor) combined with margins to account for possible microscopic tumor spread (clinical target volume), variation in the position/shape of the target region, and inaccuracies in the treatment machine and patient positioning [18]. These margins lead to a substantial amount of normal tissue inside the PTV. A good example of this is whole breast irradiation (WBI) [19], in which the entire breast is contained within the PTV and given the prescription dose. The most common side effect of WBI is breast fibrosis [20], which can have a negative impact on patients, and there have been concerns about increased toxicity with the more extremely hypofractionated schedules [21]. This makes hypofractionated WBI of potential interest for FLASH-RT: it may reduce normal tissue toxicity, such as fibrosis, and therefore facilitate hypofractionation without reducing the tumor-related outcome. WBI is also of technical interest since the PTV is large and most FLASH planning studies have focused on small target volumes [22,23]. Additionally, since WBI is a very common treatment, representing a substantial proportion of the treatments in many radiotherapy departments, facilitating more extreme hypofractionation with FLASH-RT could have a positive impact on departmental workloads. However, before clinical treatments can be developed, treatment planning considerations for FLASH-WBI, the focus of this paper, require investigation. 

Currently, WBI is delivered by two opposing tangential photon fields [24]. UHDRs are achievable on photon machines, but these are large-scale academic beamlines not suitable for clinical use [3,6]. Proton machines can also generate UHDRs [2,9], with the advantages that they are able to irradiate deep seated tissues, and, in the case of WBI, possibly only a single scanning transmission beam (TB) is necessary to achieve a comparable dose distribution to standard clinical photon plans. TBs place the Bragg-peak outside the patient and irradiate only with the proximal part of the beam, making them a practical option for proton FLASH-RT. They are also more robust against density changes and can achieve plan quality similar to volumetric-modulated arc therapy (VMAT) and close to intensity-modulated proton therapy (IMPT) for lung and head-and-neck cancer [22,23,25]. 

At the moment, many aspects of the FLASH-effect are not fully understood, but so far, it seems that not only the dose rate but also the delivered dose in a short time period are of influence. The FLASH-effect may be proportional to this dose [5], but the need for a certain minimum dose also seems likely. Different dose/fraction thresholds have been mentioned, usually in the order of 10 Gy [26], but pre-clinical research has shown an effect for doses as low as 7 Gy [7] and 4 Gy [27]. If a minimum per-fraction dose is necessary, a hypofractioned scheme would have a higher chance of meeting this requirement. The fractionation scheme for WBI has typically been 25 × 2 Gy or 15 × 2.67 Gy [18,28], but more hypofractionated regimens are now commonly used since the FAST-Forward and FAST-trials. The FAST-Forward trial compared 26 and 27 Gy in 5 daily fractions to 40 Gy in 15 fractions and concluded that both 5-fraction regimens showed non-inferiority compared to the 15 × 2.66 Gy scheme [29,30]. The FAST-trial compared the hypofractionated scheme of 5 once-weekly fractions of 5.7 or 6 Gy to 25 fractions of 2 Gy. Based on 10-year physician-assessed normal tissue effects, the 5 × 5.7 Gy and 25 × 2 Gy schedules were comparable; the 5 × 6 Gy schedule was more toxic [31]. Hypofractionated schemes are therefore supported, and their popularity has been increasing, in particular during the COVID-19 pandemic [32,33]. Although a fractional dose of 5.7 Gy is reasonably high and approaches the minimum doses typically associated with the FLASH-effect, it may be possible that more than 5.7 Gy is necessary. Therefore, other, currently unused, schemes with higher fraction doses should also be considered in analyses similar to this one [21].

This simulation study assesses the feasibility of proton FLASH for WBI by (1) evaluating the dosimetric plan quality of single-field proton TB-plans and (2) analyzing FLASH-doses for various dose and dose-rate thresholds and fractionation schemes. Additionally, since the large target volume involved in WBI may present a challenge for pencil beam scanning (PBS) FLASH-RT, (3) the ability to achieve FLASH in different treatment planning scenarios using various machine settings is also explored.

## 2. Materials and Methods

Based on recent UHDR research, a minimum dose varying from 4–10 Gy [26,27] and a minimum dose rate of 40–100 Gy/s [6,26] are necessary to produce a FLASH-effect. For most of our calculations, we have used a FLASH dose threshold T_D_ of 4 Gy and a FLASH dose rate threshold T_DR_ of 40 Gy/s. Higher dose thresholds have not been included because they will either give a similar result (when the fraction dose is higher than the threshold) or lead to no FLASH-dose (when the fraction dose is less than the threshold). A higher dose-rate threshold of 100 Gy/s was only included to compare two different calculation methods.

To be able to calculate the amount of dose delivered under FLASH conditions, a dose rate metric had to be selected. The dose rate is easy to define for single, open electron beams, but it is not so straightforward for proton spot scanning beams. Spot scanning beams have multiple spots contributing dose to a single voxel at different instantaneous dose rates and at different time points during the delivery, thereby complicating the interpretation of dose rate. Multiple dose rate calculation methods have been introduced [22,34,35,36,37] and we have selected two of these methods to calculate the amount of FLASH-dose. Both methods acknowledge that most of the voxel dose is accumulated over a short period of time since only a limited number of spots will contribute dose to a certain voxel. 

The first FLASH-dose calculation used the pencil beam scanning dose-rate (PBS-DR), which gives for each field the dose-rate in each voxel (*x*,*y*,*z*) according to the following equation [35]: $$PBS\text{-}DR\left(x,y,z\right)=\frac{D_{tot}\left(x,y,z\right)-2\cdot T_{D,PBS}}{t_{1}\left(x,y,z\right)-t_{0}\left(x,y,z\right)}$$

With D_tot_ the total voxel dose, and T_D,PBS_, a certain dose threshold. The effective irradiation time starts at t_0_, when the voxel dose surpasses the dose threshold T_D,PBS_, and ends at t_1_, when the total voxel dose minus the dose threshold T_D,PBS_, has been delivered. The total voxel dose was considered FLASH-dose if both the PBS dose rate in the voxel exceeded T_DR_ and the total voxel dose was higher than T_D_. We made our calculations using a T_D,PBS_ of 1 cGy, as previously used in the literature [23,35]. The biological meaning of this threshold is currently unknown, but it does play a crucial role in the dose-rate calculation and measurement. Especially for larger fields, an underestimation of the dose rate can occur when spots contributing only a small dose determine the starting time t_0_ and ending time t_1_ of the effective irradiation.

We have therefore also included another FLASH-dose calculation, using a recently introduced method to determine when FLASH is triggered [36,37]. This “sliding window” method considers the cumulative dose delivered as a function of time, also including dead times, and describes that a time-point *t* is considered a FLASH time-point if it lies within a time window of length T_D_/T_DR_ and the dose delivered within this window is at least T_D_. The FLASH-dose is found by summing the dose delivered at all FLASH time-points. Since this method is more computationally intensive than the first method, it is used for comparative purposes in only one of the computations. The PBS-DR was the primary calculation method. The amount of FLASH-dose is given as the average PTV-dose meeting the FLASH-requirements and is expressed as dose (Gy) or as a percentage of the average total dose.

### 2.1. Treatment Planning

The contoured CT scans of three breast cancer patients (planning target volume [PTV], 477–1047 cc), previously treated with 5-fraction WBI using two opposing tangential IMRT fields, were included. Since the FLASH-effect has been associated with a minimum dose threshold varying between 4 and 10 Gy [26,27], and as higher spot weights can provide higher instantaneous dose rates for machines with a variable current [34], schemes with a higher fraction dose (designed to maximize the amount of FLASH-effect) were also considered. Non-clinical prescriptions of 2 × 9.74 Gy and 1 × 14.32 Gy were chosen, as they are biologically equivalent to the clinical schedule of 5 × 5.7 Gy assuming α/β = 3 Gy [29]. Although non-clinical, these experimental schedules are not clinically irrelevant; they continue the clinical trend of substantially reducing the number of fractions used for breast irradiation (from a previously commonplace 25 to, currently, 5) [21].

All proton TB plans were made in the research version of the Eclipse treatment planning system (Varian Medical Systems, Palo Alto, CA, USA) and use a single TB. Since WBI is usually treated with two opposing tangential photon fields, implying that all dose-receiving tissue is treated with approximately the same dose, a single tangential TB can achieve a similar dose distribution. The tangential TBs had an energy of 250 MeV, such that the Bragg-peaks were located outside the body. They were manually placed at a beam angle between 305 and 315°, such that the field edge was aligned to the target edge and healthy tissue exposure was minimized, thereby leading to a dose distribution comparable to the clinically used two tangential IMRT fields. The plans were optimized using a minimum dose objective on the PTV and a maximum dose objective on the body.

Before creating the TB-plans for the clinical datasets, three planning scenarios with increasing complexity were performed on a representative test case (whole breast target PTV = 819 cc) in order to determine which planning parameters would give the best results.
(1)The first scenario used a single TB and placed spots of equal intensity, e.g., equal monitor units (MUs), in a uniform square grid. The spot spacing distance of the grid was varied between 4 and 10 mm to determine the grid that achieved the best trade-off between plan quality and the amount of FLASH-dose. For this simple scenario, it was expected that typically a larger portion of the dose could be delivered at UHDRs for larger spot spacing, but that plan quality would be sacrificed;(2)In the second scenario, we used a TB with a similar uniform grid and the optimal spot spacing distance obtained from the first scenario, but allowing spots to have different intensities. It was expected that most spot MUs would be high and within a narrow range, and that lower MU spots would only be necessary at the target boundary. As previous research has shown that low MU spots can substantially decrease the dose rate [23,25], different minimum MU (minMU) thresholds were used (100–800 MU), meaning that spots with fewer MUs than this threshold were removed after optimization. Similar to the previous scenario, a higher minMU threshold will likely lead to higher dose rates, but at the expense of plan quality;(3)In order to increase the amount of FLASH-dose without affecting dosimetry, we included a third scenario, which uses the concept of beam splitting described earlier for head-and-neck plans [25]. The single TB was split into two beams: one so-called FLASH-beam delivering all spots with MUs above a certain splitMU threshold and another beam delivering the remaining spots. The FLASH-beam can be delivered at a higher current as it is not limited by lower MU spots, and this will presumably lead to a higher FLASH-dose. The second beam with low MU spots was not expected to meet the FLASH-requirements but was necessary to improve plan quality. Splitting was only considered for one plan in the second scenario, namely the plan with the lowest minMU threshold of 100. This plan was expected to achieve the best quality, as more freedom in spot intensity was allowed. SplitMU thresholds of 100–1000 were evaluated (with splitMU = 100 corresponding to a non-split plan).

For the three clinical datasets, only the third scenario was planned, using minMU = 100 and the parameters deemed optimal for the test case.

In the first two scenarios, with various spot spacing distances (scenario 1) or different minMU values (scenario 2), the plans were normalized such that 2% of the PTV received 107% of the prescription dose (V107% = 2%). In the third scenario, with various splitMU values, the plan was normalized such that 98% of the target volume received 95% of the prescription dose (V95% = 98%) and the mean dose was within 100–102% of the prescription dose. In all scenarios, the FLASH-dose was calculated using the PBS-DR, and only for the clinical cases (scenario 3), the FLASH-dose was also calculated with the second “sliding window” FLASH-dose calculation method. 

### 2.2. Machine Parameters

The gantry current (GC), which determines the irradiation time and thereby the dose rate, can be varied according to two techniques. The energy-layer-based (EB) GC is constant for the entire energy layer, while the spot-based (SB) GC is adjusted for each spot. When using an EB GC, the spot with the lowest amount of MUs in an energy layer is delivered in the shortest possible irradiation time (e.g., the minimum spot time, minST). If the machine is not capable of achieving this current, it is irradiated at the highest possible current, which is the maximum nozzle current (maxN). All other spots in the energy layer are delivered with the same current, meaning that a single low-MU spot will decrease the GC and consequently the dose rate. The SB GC, adjusted per spot, delivers each spot dose in the shortest possible time or at the highest possible current. It is important to realize that a minimum MU value exists, dependent on the minST and maxN, for which there is no longer a difference between the EB GC and SB GC. This is caused by the upper limit on the current achievable by the machine: spots with MUs above this value are delivered at the highest possible current, regardless of GC technique. 

We looked at the machine settings that are currently clinically possible: minST = 2 ms and maxN = 200 nA [38,39], and at other combinations of minST and maxN that are technically feasible on certain machines: a minST of 1 ms and 0.5 ms [40,41], and a maxN of 400 nA [39]. We also included a theoretical, technically not possible, maxN of 800 nA to see if it might be useful to make such a beam in order to maximize the amount of FLASH-effect. It is of interest to see not only the influence of such a current on the amount of FLASH-dose but also on the average dose rate. 

It was assumed that the beam only irradiates when located at the spot center and is turned off during scanning, so-called discrete spot scanning. The scanning speed was chosen as 10 mm/ms, which is consistent with the literature [35]. A magnet settling time equal to the minST was added to the scanning time. The primary scanning direction was assumed to be along the smallest target dimension in the beam’s-eye view.

All assumed parameters are included in a table in the Appendix A.

## 3. Results

### 3.1. Test Case

In the first scenario, spots with equal intensity were placed in a uniform square grid with a spot spacing distance varying between 4 and 10 mm. In Figure 1, the dose volume histograms (DVHs) of all seven plans are shown, as well as the amount of FLASH-dose. Note that there is no difference in FLASH-dose between EB GC (orange/red) and SB GC (blue), due to the equal spot intensities in this scenario. The amount of FLASH-dose initially increases as spots are placed further apart, until the spot spacing distance exceeds 7 mm. PTV coverage remains sufficient for grid sizes lower than 8 mm and deteriorates when spots are placed further apart. Based on these results, we decided to select a spot spacing distance of 7 mm for the remaining planning scenarios. 

In the second scenario, plans using a single TB with a 7 mm grid were optimized with a minMU threshold of 100–800. As expected, plan quality eventually deteriorates as the minMU threshold increases (Appendix A), while the average amount of FLASH-dose generally increases for plans with a higher minMU threshold (Appendix A). For the 1 × 14.32 Gy and 2 × 9.74 Gy schemes, the absolute FLASH-dose reaches a maximum for plans with minMU = 300 and maintains these values until minMU ≥ 700. For the 5 × 5.7 Gy scheme, a similar outcome is seen for minST = 1 ms and minST = 0.5 ms, but for the clinical settings (minST = 2 ms, maxN = 200 nA, EB GC), the absolute amount of FLASH keeps increasing as minMU increases (to a maximum FLASH-dose of 3.93 Gy for minMU = 800 plans). 

Unfortunately, the plan quality of plans with minMU > 600 is insufficient, and we therefore considered the best quality plan (i.e., minMU = 100) for the third scenario, in which the beam was split into a FLASH-beam with MUs higher than a splitMU threshold and a supplementary beam with the remaining spots. Figure 2a gives the plan dose and the distribution of the spot MUs for the test case receiving 5 × 5.7 Gy, which demonstrates that most MUs are within a narrow range between 750 and 1100. The splitMU threshold was varied between 0 and 1000 and Appendix A shows some of the corresponding FLASH-doses. We see that for the 1 × 14.32 Gy and 2 × 9.74 Gy schemes, the best results are found for a splitMU ≥ 500. For the 5 × 5.7 Gy scheme, a splitMU = 700 gives the highest amount of FLASH-dose for all settings, except for the combination of minST = 2 ms and maxN = 200 nA, which reaches the highest FLASH for splitMU = 800. Overall, we can conclude that a splitMU = 700 results in the highest amount of FLASH for most fractionation schemes and machine settings, which seems to correspond with the position just below the MU peak in Figure 3a. The dose distributions of the three clinical cases and their corresponding MU histograms are shown in Figure 2b–d and show that the position of the MU peaks is comparable to that of the test case. We therefore selected a fixed splitMU of 700 for further analysis.

### 3.2. Clinical Datasets

Appendix A shows the amount of FLASH-dose for the clinical datasets, planned with splitMU = 700, calculated using the PBS-DR (a,c) and the “sliding window” method (b,d). Assuming a FLASH dose-rate threshold of 40 Gy/s, the FLASH-dose is >95% of the total dose for almost all fractionation schedules and machine settings, when calculated with the PBS-DR (Appendix A). However, for the clinically available settings of minST = 2 ms and maxN = 200 nA (for both GC techniques), 2 out of 4 cases only reach ~50%. When using the “sliding window” method (Appendix A) similar percentages are found, although the FLASH-dose is able to reach >90% in clinical settings. The difference between the PBS-DR and “sliding window” methods is more pronounced for higher FLASH dose-rate thresholds. For a FLASH dose-rate threshold of 100 Gy/s (Appendix A), we see that the PBS-DR calculation only shows a reasonable amount of FLASH-dose if the minST is low and the maxN is high, and almost no FLASH-dose for the other settings. The “sliding window” method shows at least 50% for all settings. 

Figure 3 gives the average dose rates of the four plans with a minMU = 100 and a splitMU = 700 for all fractionation schemes and machine options. We can see that the two smaller targets (Figure 3b,d) achieve higher average dose-rates than the two larger targets (Figure 3a,c). For the clinical settings (minST = 2 ms, maxN = 200 nA), the larger targets have an average dose rate only slightly higher than 40 Gy/s, while the smaller targets achieve >50 Gy/s. Average dose rates of ≥100 Gy/s are only reached under certain conditions (higher fractionation doses, lower minST, and higher maxN). 

## 4. Discussion

In this work on TB proton FLASH for WBI we investigated how the amount of FLASH and dose rates can be improved by adjusting treatment planning parameters and machine settings. WBI using a single TB is technically feasible, as the TBs will be expected to be similar, but likely larger, than in FAST-01 in which bone metastases are treated with a single TB delivering 8 Gy [16], and we have now shown that acceptable dose distributions can be achieved for this single TB framework. The amount of FLASH estimated is strongly dependent on the dose-rate calculation method, and the machine parameters. For the current machine parameters (5 × 5.7 Gy; minST = 2 ms, maxN = 200 nA, EB GC), the amount of FLASH-dose is limited, but beam-splitting can partly overcome this issue, making WBI a serious candidate for FLASH-RT.

We found that without beam-splitting a maximum FLASH-percentage of 71% was achieved for the test case, for the current clinical framework and a minMU of 800. Unfortunately, for this, minMU the PTV is underdosed (V95% = 78.9%) but decreasing the minMU results in less FLASH (39% for minMU = 600 to 1% for minMU = 100). The trade-off between plan quality and FLASH-dose remains a challenge: low MU spots are necessary to achieve plan homogeneity, but also decrease the amount of FLASH-dose. The influence of low MU spots can be reduced by splitting the beam into a FLASH-beam, delivered at a high current, and a residual beam, which delivers the low MU spots at a lower current. Beam-splitting resulted in ~50% (test case, patient 2) or >95% (patient 1,3) FLASH for the clinical settings of minST = 2 ms and maxN = 200 nA, when the dose rate is calculated using the PBS-DR (Appendix A). This percentage increased to ~90% FLASH for the test case and patient 2 (and stayed at >95% for patient 1,3), when calculated with the “sliding window” method. The size of the target seems to have a substantial effect on the amount of FLASH when using the PBS-DR. Patient 1 and 3, both achieving >95% FLASH, have PTVs of 677 cc and 477 cc, respectively, while patient 2 and the test case, only achieving ~50% FLASH, have PTVs of 1047 cc and 819 cc, respectively. We can conclude that beam-splitting allows for more FLASH in all cases, but that the best results are obtained for smaller targets. For the “sliding window” method the size of the target does not seem to play such an important role. 

Another noteworthy observation is that if the FLASH dose-rate threshold is 40 Gy/s, a FLASH-percentage of >95% can be achieved by adjusting only one of the following parameters: decrease minST to 1 ms, increase maxN to 400 nA, or select a fractionation scheme of 2 × 9.74 Gy. Using an SB GC instead of an EB GC does not make a difference, since only the irradiation time of spots with an MU below a certain value are affected by this change and the splitMU-threshold of 700 exceeds this value. Increasing the maxN from 400 nA to 800 nA only has a limited effect (Appendix A): for a FLASH dose threshold of 40 Gy/s the amount of FLASH-dose is comparable between maxN = 400 nA and 800 nA; for a FLASH dose threshold of 100 Gy/s there is an increase in FLASH dose, in particular when using the PBS-DR calculation method. More insight into the dose threshold and relevant dose rate definition will therefore decide whether it is useful to increase the nozzle current of a beam.

Figure 3 shows that, as expected, the smaller targets achieve the highest dose-rates and that for a minST = 2 ms/maxN = 200 nA the two largest targets (Figure 3a,c) have an average dose-rate only slightly larger higher than 40 Gy/s, which was selected as the FLASH dose-rate threshold T_DR_. This might explain why the FLASH-percentages are reasonably low for these two cases. The exact value of T_DR_, the required minimum dose rate to produce a FLASH-effect, is currently unclear, and multiple FLASH dose-rate thresholds have been mentioned. From Figure 3 we can conclude that if the dose rate threshold turns out to be around ~100 Gy/s, improvements to the machine (lower minST, higher maxN) and/or regimens with fewer fractions are necessary to achieve these dose-rates. 

We only considered the PBS-DR and the “sliding window” method for our calculations, but other methods, such as the dose-averaged DR (DADR) [34] and the UHDR-contribution (UHDRc) based on spot DRs [22], have also been described. It is unclear which of these are most biologically relevant, and they all focus on different aspects of delivery: the DADR and UHDRc do not take scanning time into account and are therefore more biased towards the instantaneous dose rate, while the “sliding window” and PBS-DR are biased towards the average dose rate. However, their parameters can be adjusted to skew more towards the instantaneous dose rate. The dose threshold T_D,PBS_, used by the PBS-DR, is selected fairly arbitrarily and might lead to the inclusion of spots far away. Appendix A shows that by increasing T_D,PBS_ to 10 cGy, the amount of FLASH increases slightly to ~65–70% for the larger targets (minST = 0.5 ms, maxN = 200 nA, EB GC, 5 × 5.7 Gy). Note that the “sliding window” method dose not yield a dose-rate, but rather a FLASH-dose, and although it seems to make more sense intuitively, it also strongly depends on the choice of FLASH-thresholds T_D_ and T_DR_. By doubling T_D_ (while keeping T_DR_ constant), the time window doubles, thereby increasing the possibility of achieving FLASH. The method also allows for theoretical “re-triggering” of FLASH in the same voxel, which is currently not supported by any biological evidence. 

In Appendix A the results of the PBS-DR and the “sliding window” method are given for T_DR_ = 40 Gy/s and they show that the “sliding window” method only achieves a higher FLASH-percentage for minST = 2 ms, maxN = 200 nA and 5 × 5.7 Gy fractionation scheme, all other settings achieve a comparable or lower amount of FLASH. In particular a higher fractionation dose results in lower FLASH, which can be explained by the increased irradiation time due to the higher fraction dose: more time passes between different spot contributions to a certain voxel and therefore some spot-contributions may fall outside the time window (which is 100 ms for T_D_ = 4 Gy and T_DR_ = 40 Gy/s). This is clarified by Appendix A, in which the spot contributions to a voxel are given for the 5 × 5.7 Gy and 1 × 14.32 Gy schemes. For a higher FLASH dose-rate threshold the difference between PBS-DR and “sliding window” method is much larger (Appendix A) and it indicates that the arbitrary PBS dose threshold T_D,PBS_ has a negative impact on the estimated amount of FLASH. However, by adjusting T_D,PBS_ the difference between these methods can be eliminated and it is important to realize that at this moment all methods are based on assumptions. Future research is necessary to determine which dose rate method is biologically meaningful. Similarly, although our definition of FLASH-dose is based on parameters supported by biological FLASH research, it remains an uncertain measure and does not prove that a FLASH-effect will occur for this dose. To establish the actual FLASH-dose and effect more translational and clinical research is needed.

Another assumption in need of more verification is the belief that the two split beams can be considered as separate beams not influencing each other’s biological dose. Assuming oxygenation depletion is the mechanism behind the FLASH-effect, it is plausible to assume that the environment has been restored before the next irradiation. However, other theories to (partially) explain the FLASH-effect have also been proposed and would not necessarily support this assumption. Therefore, the minimum time between deliveries necessary to consider them as separate events still needs to be determined. It is also important to note that robustness will need to be included as the time between the deliveries of the split beams increases.

Most pre-clinical proton FLASH experiments, using both passive scattering [9,11] and PBS [12,13] as a delivery method, have used TBs. However, the FLASH-effect has recently been observed in mouse abdominal tissues for passive scattering spread-out Bragg-peak irradiation [10] and in mouse brains after Bragg-peaks PBS [8]. Most proton FLASH treatment planning research has also used TBs [22,23,25,34], but Bragg-peak planning has recently gained more interest [36,42]. Although the precise positioning of the Bragg-peak and the absence of exit dose are the biggest advantages of proton radiotherapy, TBs are used in a currently ongoing FLASH-trial because of their ability to achieve sufficient plan quality in a more practical manner [16]. TBs are also very robust for possible changes in breast size and shape.

WBI was selected for several reasons: (1) the breast allows for the use of a single TB, also eliminating the uncertainties concerning the unknown consequences of multiple beam irradiation on the FLASH-effect; (2) the hypofractionated treatment has a large patient population [43], meaning that even if a small fraction experiences the benefits of UHDR therapy. This can still be a large group of patients; and (3) the PTV of WBI consists mostly of healthy tissue, therefore a FLASH-effect could be reasoned to have consequences for the amount of toxicity. 

We would also like to point out that further enhancements can be obtained by improving the treatment planning process. For example, the FLASH-beam and the remaining beam could be optimized simultaneously: the FLASH-beam should be optimized such that the highest amount of FLASH is achieved, while the other beam should be optimized using a finer dose grid to enhance plan quality.

An important point of concern regarding proton TB radiotherapy is the skin dose. Photon beam irradiation has skin-sparing properties when entering the body [44] and, to a lesser extent, before exiting the body [45,46]. For a 250 MeV proton beam only a modest skin-sparing effect exists [47]. When using proton TBs, the skin receives almost the prescription dose, which is generally too high and if there is insufficient FLASH-effect, this will limit clinical possibilities to tumors in which a high skin dose is permissible [48]. Treating other breast cancers with FLASH-WBI (particularly with double and single fraction schemes) will presumably only be acceptable after a FLASH-effect has been confirmed in the skin and no excess toxicity has been observed. If that is the case, then one could imagine that FLASH-WBI might be initiated by studies that initially treat only 1 fraction of a 5-fraction scheme with a FLASH-TB. With acceptable outcomes, the FLASH-TB component could then be increased to 2–5 fractions. The 1 and 2 fractionation schemes used in this work are for investigative purposes only. They would currently not be considered acceptable using photon therapy, but a potential FLASH-effect might reduce the biological dose in normal tissues to an acceptable level. For example, if there was a FLASH sparing effect of ~40%, the 2 × 9.74 Gy scheme would biologically translate to ~2 × 6 Gy, and the single fraction of 14.32 Gy would be comparable to a biological dose of ~10 Gy, which could be delivered without expectations of excessive toxicity. 

A few possible concerns for clinical application of FLASH for WBI have to be expressed. Firstly, up till now, FLASH experiments have mostly demonstrated a FLASH effect on healthy tissues and absence of a FLASH effect for macroscopic tumors. However, in the context of adjuvant WBI, we are dealing with a PTV free of macroscopic tumor, and it is currently unclear how UHDR-irradiation affects microscopic cancer. If there is a sparing effect for microscopic tumor cells similar to that for healthy tissue, using FLASH-RT will be harmful rather than helpful for WBI and for many other indications. Next, FLASH effects have been mostly observed in small volumes. There is no proof yet that a FLASH effect can exist in a large volume such as a WBI. Finally, not much is known yet about the long term effects of UHDR irradiation. One recent publication on cats and a mini-pig [14] reported that the experiment on cats was prematurely interrupted due to maxillary bone necrosis occurring 9–15 months after FLASH-RT, and that the mini-pig also showed severe late skin necrosis at 7–9 months post-FLASH-RT [49]. They highlight that it is necessary to proceed with caution and collect more information on long-term effects of UHDR irradiation as many of the women treated with WBI have a long survival and lack of microscopic tumor control or more late toxicities can impact for many years after treatment.

## 5. Conclusions

We demonstrated that using a single UHDR TB for WBI can achieve acceptable plan quality and a high amount of FLASH-dose. For a FLASH dose rate threshold of 40 Gy/s a FLASH-dose percentage of 50–95% can be achieved for currently used machine parameters (minST = 2 ms, maxN = 200 nA, EB GC). For a higher FLASH dose-rate threshold of 100 Gy/s more favorable machine parameters (shorter minST, higher maxN, SB GC) or beam splitting are necessary for a reasonable amount of FLASH-dose, and the sliding window method for calculating dose-rate predicts more FLASH than the pencil beam scanning dose rate. Hypofractionation in five fractions is clinically very common and a potential FLASH-effect might in the future lead to shorter fractionation schemes. Additionally, the single TB setup is currently used in a clinical trial, meaning that the clinical framework is already present. Based on our results we can conclude that WBI makes a very suitable candidate for FLASH radiotherapy.

## Figures and Tables

**Figure 1 cancers-15-02579-f001:**
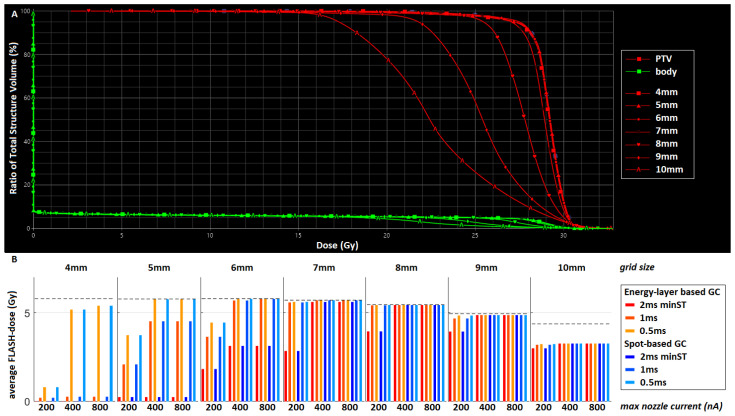
(**A**) Dose-volume histogram of the PTV (red) and body (green) for the plans with varying grid spacing (4–10 mm). PTV DVHs for 6–10 mm show increasing coverage, while for 4–6 mm grid spacing DVHs are the same with the best target coverage; (**B**) average absolute FLASH-dose (Gy) for the different grid spacing plans and machine settings. The bars in red correspond to an EBGC and those in blue to an SBGC; the darkest bars correspond to a minST = 2 ms, the medium-dark bars to a minST = 1 ms, and the lightest bars to a minST = 0.5 ms; for each grid spacing, the FLASH-dose is given for a maxN = 200 nA (left), 400 nA (middle), and 800 nA (right). All these results are for the 5 × 5.7 scheme; use the PBS-DR with a threshold of 1 cGy and assume FLASH-thresholds of TD = 4 Gy and TDR = 40 Gy/s.

**Figure 2 cancers-15-02579-f002:**
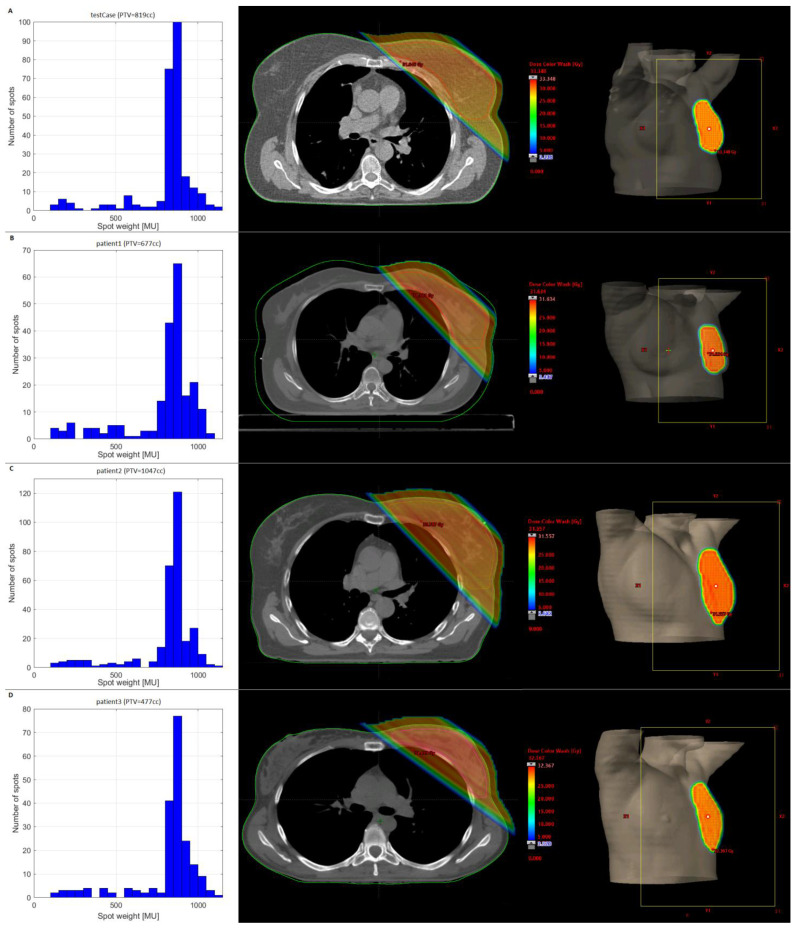
Histogram of spot-MU distribution (left), dose distribution (middle), and beam-eye’s view (right) for the (**A**) test case and (**B**–**D**) three clinical datasets.

**Figure 3 cancers-15-02579-f003:**
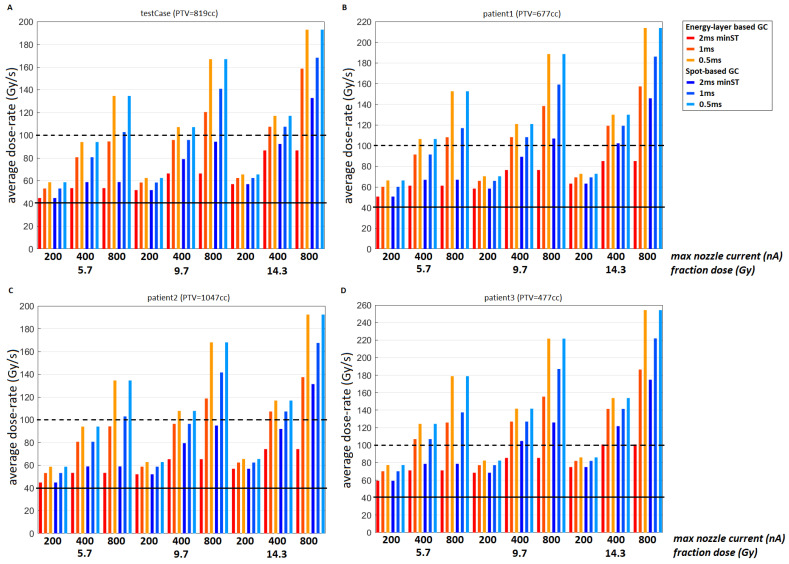
Average dose rate (Gy/s) for the split plan (minMU = 100, splitMU = 700) of the (**A**) test case and (**B**–**D**) three clinical cases. The bars in red correspond to an EB GC and those in blue to an SB GC; the darkest bars correspond to a minST = 2 ms, the medium-dark bars to a minST = 1 ms, and the lightest bars to a minST = 0.5 ms; for each case, the results are given for all three fractionation doses (5.7 Gy, 9.7 Gy, and 14.3 Gy), and per fractionation scheme, the dose rate is given for a maxN = 200 nA (left), 400 nA (middle), and 800 nA (right). All these results use the PBS-DR with a threshold of 1 cGy. The solid horizontal line denotes a dose rate of 40 Gy/s, and the dashed line a dose rate of 100 Gy/s.

## Data Availability

We do not have approval to share research data or transfer it outside the institution.

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
