# Peer review of "Single Ultra-High Dose Rate Proton Transmission Beam for Whole Breast FLASH-Irradiation: Quantification of FLASH-Dose and Relation with Beam Parameters"

_cancers, 2023, doi:10.3390/cancers15092579_

Round 1

Reviewer 1 Report

The authors studied whole breast irradiation plan quality and determined FLASH-dose for various machine settings using ultra-high dose rate proton transmission beams. The authors have reported similar study for head and neck cancer previously. Although treatment planning for whole breast irradiation is simpler than for head and neck, challenging is much larger volume of the target. In addition to splitting beam, in this study the authors employed sliding window” method for clinical data sets. I think the topic is of interest. This study is well-designed, the methodology is scientifically sound. The manuscript in general is well-structured and most issues are explained in enough detail. I think that the manuscripts can be accepted for the publication after considering following minor issues.

Abstract and Conclusion

Please refer “sliding window” method in Abstract and Conclusion because in this study it is key issue for dosimetric evaluation of ultra-high dose rate irradiation.

Lines in dose-volume histogram in figure 1 and 2 cannot be identified by symbols. Please rework them.

Small characters in figure 5 cannot be identified. Please rework them.

4. Discussion

The last paragraph starting with “Two things are important” is better to be omitted because there is no direct relationship with the main subject of this study.

References 36

“Med Phys 2021;1-13” should be “Med Phys 2022; 49(3):2026-2028”

Reviewer 2 Report

There was a lot of work done here but it was all based on assumptions and very little on data. It was as if the desire to show you were first was more important than thinking through the issues. 

1. Do we even know if FLASH can work across a volume like the human breast - anywhere. The work in humans would nto show failure, at least not the bone work, because 8 Gy x 1 is super well tolerated.

2. Fractionated radation for breast is well tolerated. Do we actually need to get better sparing. Even UK/Canadian fractionation schedules are well tolerated. At what point is theoretical convenience worth treating women potentially very poorly...if these large fractions done spare normal tissue you will be guaranteeing great toxlicity. Look at the old HDR data and the soft tissue shrinkage that results causing very real cosmetic issues. 

3. Does the "split beam" get you FLASh or does the slower RT stuff cause a loss of FLASH benefit. You propose a solution without data.

4. To say this can be done in a patient with inflammatory breast cancer suggests you don't understand the issues with volume...it would really need Bragg peak therapy in a complex shape making more than one field angle likely to be needed, likely abrogating FLASH. This should be removed.

5. Saying you can do the tech is one thing...the plots can be taken offline and put into supplementary data save the first one. The subesequent ones just fill up the paper with data that, given the above, may not mean much.

How will you confirm dose delivery with dosimetry?

How will you stop a treatment that is not going per plan?

How will peple be able to know how this was done and optimized downstream?

How robust is this approach in a breathing patient or in day to day set up variation given breasts swell, etc. 

Round 2

Reviewer 2 Report

With the additional comments which basically say we have no idea if this is a good idea, if it could work, or if it could actually hurt patients - recent large volume data suggest that there may be serious late effects, then it is ok to publish as a thought experiment only.

Much more data are needed before we deploy such things.